# LEARNING TO SAMPLE WITH ADVERSARIALLY LEARNED LIKELIHOOD-RATIO

**Chunyuan Li, Jianqiao Li, Guoyin Wang & Lawrence Carin**
Duke University
{chunyuan.li,jianqiao.li,guoyin.wang,lcarin}@duke.edu

## ABSTRACT

We link the reverse KL divergence with adversarial learning. This insight enables learning to synthesize realistic samples in two settings: ($i$) Given a set of samples from the true distribution, an adversarially learned likelihood-ratio and a new entropy bound are used to learn a GAN model, that improves synthesized sample quality relative to previous GAN variants. ($ii$) Given an unnormalized distribution, a reference-based framework is proposed to learn to draw samples, naturally yielding an adversarial scheme to amortize MCMC/SVGD samples. Experimental results show the improved performance of the derived algorithms.

## 1 BACKGROUND ON THE REVERSE KL DIVERGENCE

**Target Distribution** Assume we are given a set of samples $\mathcal{D} = \{x_i\}_{i=1,N}$, with each sample assumed drawn iid from an unknown distribution $q(x)$. For $x \in \mathcal{X}$, let $\mathcal{S}_q \subset \mathcal{X}$ represent the support of $q$, implying that $\mathcal{S}_q$ is the smallest subset of $\mathcal{X}$ for which $\int_{\mathcal{S}_q} q(x)dx = 1$ (or $\int_{\mathcal{S}_q} q(x)dx = 1 - \epsilon$, for $\epsilon \to 0^+$). Let $\mathcal{S}_q^o$ represent the complement set of $\mathcal{S}_q$, $i.e.$, $\mathcal{S}_q \cup \mathcal{S}_q^o = \mathcal{X}$ and $\mathcal{S}_q \cap \mathcal{S}_q^o = \emptyset$.

**Model Distribution** We desire a model $p_\theta(x)$ that approximately allows one to draw samples efficiently from $q(x)$, implemented as $x = f_\theta(z)$ with $z \sim p(z)$, where $p(z)$ is a distribution that one may sample from easily, and $f_\theta(z)$ is a nonlinear deterministic function with parameters $\theta$ that are to be learned. Similarly, let $\mathcal{S}_{p_\theta}$ represent the support of $p_\theta$, with $\mathcal{S}_{p_\theta} \cup \mathcal{S}_{p_\theta}^o = \mathcal{X}$ and $\mathcal{S}_{p_\theta} \cap \mathcal{S}_{p_\theta}^o = \emptyset$.

The reverse KL divergence [1] between these two distributions is:

$$\text{KL}(p_\theta(x)||q(x)) = \mathbb{E}_{p_\theta(x)} \log \left[\frac{p_\theta(x)}{q(x)}\right] = -h(p_\theta(x)) - \mathbb{E}_{p_\theta(x)} \log q(x) \tag{1}$$

- The 1st term is the differential entropy, encouraging $p_\theta(x)$ to spread over the support set as wide as possible
- The 2nd term can be further written as: $\mathbb{E}_{p_\theta(x)} \log q(x) = \int_{\mathcal{S}_{p_\theta} \cap \mathcal{S}_q} p_\theta(x) \log q(x)dx + \int_{\mathcal{S}_{p_\theta} \cap \mathcal{S}_q^o} p_\theta(x) \log q(x)dx$, where there is a strong (negative) penalty introduced by $\int_{\mathcal{S}_{p_\theta} \cap \mathcal{S}_q^o} p_\theta(x) \log q(x)dx$. Hence, it is encouraged that $\mathcal{S}_{p_\theta} \cap \mathcal{S}_q^o = \emptyset$, implying $\mathcal{S}_{p_\theta} \subseteq \mathcal{S}_q$. When $\mathcal{S}_{p_\theta} \subset \mathcal{S}_q$, "mode-collapse" is manifested.

It can be seen that the goals of two terms in the reverse KL objective seem complementary to each other. We advocate that minimizing $\text{KL}(p_\theta||q)$ is a promising approach to learn a model $p_\theta(x)$ to characterize $q(x)$. Below, we discuss two distinctive setups to learn $p_\theta(x)$, when only either a sample set (in Section 2) or an unnormalized density form (in Section 3) of $q(x)$ is available.

## 2 LEARNING WITH SAMPLES

Learning $p_\theta(x)$ with a set of samples from $q(x)$ is exactly the problem setup of Generative Adversarial Networks (GAN) (Goodfellow et al., 2014). One may consider to train a $\psi$-parameterized discriminator $g_\psi(x)$ to estimate the likelihood-ratio $\log(p_\theta(x)/q(x))$ (Kanamori et al., 2010; Mohamed & Lakshminarayanan, 2016; Mescheder et al., 2016; Gutmann & Hyvärinen, 2010):

$$\hat{\psi} = \text{argmax}_\psi [\mathbb{E}_{p(z)} \log \sigma(g_\psi(f_\theta(z))) + \mathbb{E}_{q(x)} \log(1 - \sigma(g_\psi(x)))], \tag{2}$$

---

[1] In contrast to the maximum likelihood setup $\text{KL}(q(x)||p_\theta(x))$

where $\sigma(x) = 1/(1 + \exp(-x))$, and with both expectations in (2) implemented approximately via samples. One may show that for fixed $\theta$, the optimal solution to (2) is $g_{\hat{\psi}}(x) = \log[p_\theta(x)/q(x)]$. Utilizing $g_{\hat{\psi}}(x)$ as an approximation for the likelihood-ratio, we seek to minimize $\mathrm{KL}(p_\theta\|q)$ via

$$\hat{\theta} = \mathrm{argmin}_\theta \, \mathbb{E}_{p(z)} g_{\hat{\psi}}(f_\theta(z)) \tag{3}$$

Learning proceeds by alternating between updating $\psi$ (defining the discriminator/critic) and $\theta$ (defining the generator/actor). We call this procedure as ALL: *Adversarially Learned Likelihood-ratio*. It reveals the close connections to standard GAN training (Goodfellow et al., 2014), except two minor differences detailed in the Supplementary Material (SM).

Furthermore, the reverse KL objective in (1) has no control on the relative importance of the two terms; we may rather minimize $-\lambda h(p_\theta) - \mathbb{E}_{p_\theta} \log q(x)$, with $\lambda \geq 1$ allowing control of how much emphasis is placed on mitigating mode collapse. To achieve this we require a means of estimating or bounding $h(p_\theta)$ using samples from $p_\theta(x)$.

**Lemma 1** *Let $t_\xi(z|x)$ be an auxiliary encoder associated with the generative model $p_\theta(x)$, for which $x = f_\theta(z)$ and $z \sim p(z)$. The mutual information between $x$ and $z$ satisfies*

$$I(x; z) = h(p_\theta(x)) \geq h(p(z)) + \mathbb{E}_{p(z)} \log t_\xi(z|f_\theta(z)).$$

The proof is provided in the SM. Since $h(p(z))$ is a constant wrt $(\theta, \xi)$, one may seek to maximize $\mathbb{E}_{p(z)} \log t_\xi(z|x)$ to increase the $h(p_\theta)$. We therefore modify the learning of $\theta$ in (3)as:

$$\hat{\theta} = \mathrm{argmin}_\theta \, \mathbb{E}_{p(z)}[g_{\hat{\psi}}(f_\theta(z)) + (1 - \lambda) \, \log t_\xi(z|f_\theta(z))] \tag{4}$$

with $\lambda \geq 1$, controlling the impact of the entropy regularizer. Meanwhile, $t_\xi(z|x)$ is also optmized to satisfy the cycle-consistency of $z$ (Li et al., 2017a; Zhu et al., 2017). Hence, the learning is referred to as ALLEN: *Adversarially Learned Likelihood-ratio with ENtropy*.

## 3 LEARNING WITH AN UNNORMALIZED DISTRIBUTION

Assume $q(x) = u(x)/C$, with $u(x)$ known and $C = \int u(x)dx$ unknown. Our goal is to design a model $p_\theta(x)$ to sample from $q(x)$, based on $u(x)$ and *without* access to samples from $q(x)$. Therefore, we seek to minimize $\mathbb{E}_{p_\theta(x)} \log \left[\frac{p_\theta(x)}{u(x)}\right] = \mathrm{KL}(p_\theta\|q) - \log C$ with the following two methods.

### 3.1 A REFERENCE-BASED SAMPLING FRAMEWORK

We can re-write the reverse KL divergence with a reference distribution $r(x)$ as:

$$\mathrm{KL}(p_\theta(x)\|u(x)) = \mathbb{E}_{p_\theta(x)} \left[\log \frac{p_\theta(x)}{r(x)} + \log \frac{r(x)}{u(x)}\right], \tag{5}$$

where $r(x)$ should satisfy two minimum requirements: ($i$) a sampling mechanism to draw samples, and ($ii$) a functional form to evaluate density. Therefore, by leveraging the corresponding ability, the 1st term in the expectation can be implicitly estimated via the ALLEN in (2)(4) using the two sample sets, and the 2nd term can be explicitly evaluated using their density forms. In practice, we have a wide family of $r(x)$ to choose, including Gaussian distributions, mixture of Gaussians, and normalizing flows Rezende et al. (2015); Kingma et al. (2016) *etc*. From this perspective, Adversarial Variational Bayes (AVB) (Mescheder et al., 2016) can be viewed as a special case when employing the prior or empirical Gaussian as the reference distributions in the Bayesian setup.

### 3.2 ADVERSARIAL SAMPLE AMORTIZATION

The alternative approach is to directly draw samples from $u(x)$ using Markov Chain Monte Carlo (MCMC) (Brooks et al., 2011; Welling & Teh, 2011) or Stein Variational Gradient Descent (SVGD) (Liu & Wang, 2016) methods. However, these methods become inefficient when we need to apply them repeatedly on a large number of different but similar target distributions, because one has to start the dynamics from the scratch and wait until convergence. This problem can be addressed via sample amortization (Feng et al., 2017; Li et al., 2017b): training a model $p_\theta(x)$ such as a stochastic neural network to approximate the samples of the multiple distributions. In learning, $\theta$ is iteratively adjusted to approach the incrementally improved samples along the dynamics, and finally move towards the target distribution. Our reverse KL framework naturally leads to fitting samples between the model and the dynamics via ALLEN in (2)(4), in contrast to the $\ell_2$ metric in (Feng et al., 2017). It has been shown that the adversarial schemes can match the marginal/conditional distributions (Goodfellow et al., 2014; Li et al., 2017a). Therefore, the samples from dynamics can approximate the target distribution at convergence, so as the samples drawn from our model $p_\theta(x)$.

## 4 EXPERIMENTAL RESULTS

### 4.1 LEARNING WITH SAMPLES

We first study the capability of different models in terms of mode coverage/separation. Following the design in (Metz et al., 2017; Nguyen et al., 2017), we consider a synthetic dataset of samples drawn from a 2D mixture of 8 Gaussian distributions. The ALL is first compared with the original GAN and three state-of-the-art GAN variants: Spectral Normalization (SN)-GAN (Miyato et al., 2018), Unrolled-GAN (Metz et al., 2017) and D2GAN (Nguyen et al., 2017). For all variants, we also study their entropy-regularized versions of all methods, by simply adding the entropy bound ($\lambda = 2$ by default) proposed in (4), when training the generator.

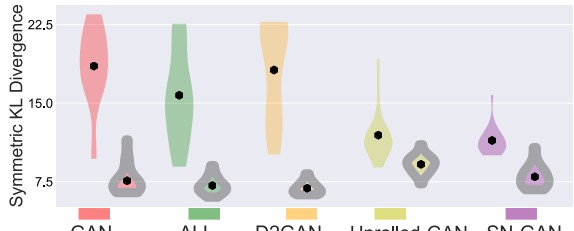

Twenty runs were conducted for each algorithm. Since we know the true distribution in this case, we employ the symmetric KL divergence as a metric to quantitatively compare the quality of generated data. In Fig. 1 we report the distribution of divergence values for all runs. The ALL variant improve over the original GAN, because the former may ease the gradient-vanishing issue of the latter (Arjovsky & Bottou, 2017; Pu et al., 2018). We add the entropy regularizer to each variant, and visualize their results as violin plots with black edges (the color for each variant remains for comparison).

Figure 1: Comparison of different GAN variants. The GAN models and corresponding entropy-regularized variants are visualized in the same color; in each case, the left result is unregularized, and the right employs entropy regularization. The black dots indicate the means of the distributions.

The largely decreased divergence mean and reduced variance show that the entropy regularizer yields significantly more consistent and reliable solutions, across all methods.

### 4.2 LEARNING WITH AN UNNORMALIZED DISTRIBUTION

To provide insight into the differences of the representation power between various sampling/amortization methods, we consider two unnormalized 2D densities $p(\boldsymbol{z}) \propto \exp[-U(\boldsymbol{z})]$. Fig. 2 shows several analyses: ($i$) The first pair compare the methods related to AVB (Mescheder et al., 2016), here considering two forms of reference distributions, including empirical Gaussian and based on normalizing flows (NF); By comparing plots in Fig. 2(b) and (c), we see a substantial improvement in the approximation quality based on use of a flexible reference distribution (here NF). ($ii$) The second pair of results consider two methods for amortizing the output of SVGD; Previous method employed the $\ell_2$ distance (Feng et al., 2017). We propose the adversarial scheme described in Sec.3.2 to amortize the samples. The comparison of methods for amortizing based on SVGD samples is presented in plots Fig. 2(d) and (e). ($iii$) Similarly, the final pair of results examine two methods for amortizing samples from an Langevin-based MCMC sampler, shown in Fig. 2(f) and (g). We observe that ALLEN accurately approximates the entire distributions, while the $\ell_2$-based methods tend to collapse to the mean of the distribution; this is because the optimum of an $\ell_2$ objective is the mean.

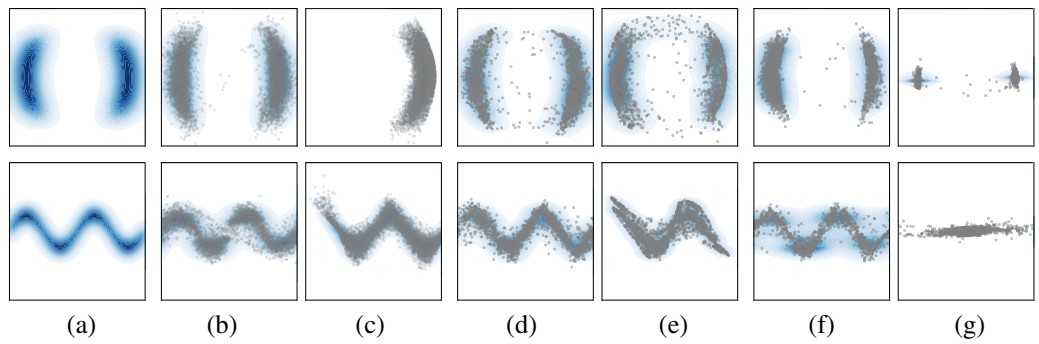

|  (a)  |  (b)  |  (c)  |  (d)  |  (e)  |  (f)  |  (g)  |

Figure 2: Comparison on learning with unnormalized distributions.

Learning to sample from unnormalized distributions is quite useful in many interesting applications. We show the results for Variational Autoencoders (Kingma & Welling, 2014) in SM, and point out future works, including Bayes by HyperNet (Pawlowski et al., 2017) and Soft Q-learning (Haarnoja et al., 2017) *etc*.

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

# A  FROM THE ORIGINAL GAN TO ALLEN

We describe the differences between original GAN and ALLEN as the following:

**Discriminator**  In the original GAN (Goodfellow et al., 2014), one may consider to train a $\psi$-parameterized discriminator $g_\psi(x)$:

$$\hat{\psi} = \text{argmax}_\psi[\mathbb{E}_{q(x)} \log \sigma(g_\psi(x)) + \mathbb{E}_{p(z)} \log(1 - \sigma(g_\psi(f_\theta(z))))], \tag{6}$$

where $\sigma(x) = 1/(1 + \exp(-x))$. This is similar to (2) in the main paper, except that the two expectation terms are switched.

**Generator**  To train the generator, original GAN minimizes:

$$\hat{\theta} = \text{argmin}_\theta \mathbb{E}_{p(z)} \log(1 - \sigma(g_\psi(f_\theta(z)))), \tag{7}$$

or the alternative "log D trick":

$$\hat{\theta} = \text{argmin}_\theta \mathbb{E}_{p(z)} - \log(\sigma(g_\psi(f_\theta(z)))). \tag{8}$$

(7) and (8) are similar to (3) in the main text, but different in the presence of the sigmoid function. According to Arjovsky & Bottou (2017), optimization of (7) and (8) lead to gradient vanishing or unstable issues.

**Entropy Bound**  After carefully implementing the two above small differences, the major difference of ALLEN is the use of cycle-consistency to bound the entropy, as implemented in (4).

# B  PROOF OF THE ENTROPY BOUND IN LEMMA 1

Consider random variables $(x, z)$ under the joint distribution $p_\theta(x, z) = p(z)p_\theta(x|z)$, where $p_\theta(x|z) = \delta(x - f_\theta(z))$. The mutual information between $x$ and $z$ satisfies $\text{I}(x; z) = h(x) - h(x|z) = h(z) - h(z|x)$. Since $p_\theta(x|z)$ is a deterministic function of $z$, $h(x|z) = 0$. We therefore have $h(x) = h(z) - h(z|x)$, where $h(z) = -\int p(z) \log p(z) dz$ is a constant wrt $\theta$. For general distribution $t_\xi(z|x)$,

$$h(z|x) = -\mathbb{E}_{p_\theta(x,z)} \log p_\theta(z|x) \tag{9}$$
$$= -\mathbb{E}_{p_\theta(x,z)} \log t_\xi(z|x) - \mathbb{E}_{p_\theta(x)} \text{KL}(p_\theta(z|x) \| t_\xi(z|x))$$
$$\leq -\mathbb{E}_{p_\theta(x,z)} \log t_\xi(z|x) \tag{10}$$

We consequently have

$$h(x) = -\mathbb{E}_{p_\theta(x)} \log p_\theta(x) dx$$
$$= h(z) - h(z|x) \geq h(z) + \mathbb{E}_{p_\theta(x,z)} \log t_\xi(z|x) \tag{11}$$

Therefore, entropy is upper bounded by the negative likelihood or cycle-consistency loss; minimizing the cycle-consistency loss maximizes the entropy or mutual information. $\square$

# C  RESULTS ON VARIATIONAL AUTOENCODERS

We consider sampling the posterior distribution of the latent code in VAE (Kingma & Welling, 2014). The prior is $\mathcal{N}(\mathbf{0}, \mathbf{I})$. To illustrate the difference, we trained the VAE on a synthetic dataset similar to Fig. 5 in (Mescheder et al., 2016). This example contains 10 data points, each a 10D one-hot vector, with non-zero element at the different positions, and the latent space is 2D.

We visualize the resulting division of the latent space in Fig. 3, where each color corresponds to the conditioning observation in the $x$-space: $q_\phi(z) = \int q_\phi(z|x)q(x)dx$. Ideally, the entire dot cloud should look like a standard normal distribution, and each individual component should be well separated. The standard VAE learns the mixtures of Gaussian, due to the variational assumption. The standard AVB can be viewed as employing the prior as the reference, which can learn more complex distribution for each component. However, the boundary is not well separated. When

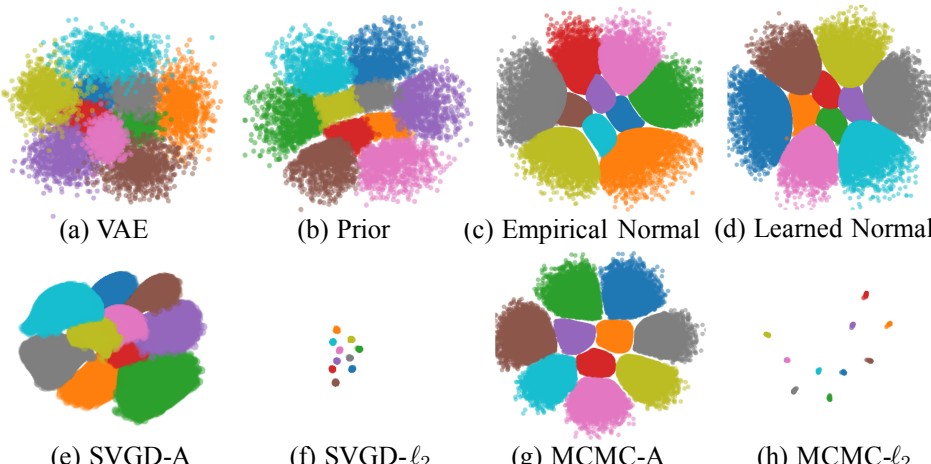

(a) VAE     (b) Prior     (c) Empirical Normal   (d) Learned Normal

(e) SVGD-A     (f) SVGD-$\ell_2$     (g) MCMC-A     (h) MCMC-$\ell_2$

Figure 3: Comparison of latent spaces; the colors mean conditioning on different observations. From our reference-based framework, (b)-(d) chooses different references. (e)-(f) are results of amortizing SVGD/MCMC using ALL and $\ell_2$, respectively.

choosing more flexible empirical Gaussian references, we see substantial improvement. We propose to use the learned normal-parameterized encoder in standard VAE as the reference, and the results are visualized in Fig. 3(d). It learns both the correct overall shape and a clear boundaries. Interestingly, the empirical normal proposed in the adaptive constrast technique (Mescheder et al., 2016) seems to provide visually similar latent distributions. When comparing the amortizing SVGD/MCMC methods, ALLEN performs better at maintaining the distributions, while the $\ell_2$ method collapses to the mode of each distribution.

