# OpenReview forum: "Learning to Sample with Adversarially Learned Likelihood-Ratio"
_ICLR.cc/2018/Workshop — Reject_

### Official Review · AnonReviewer3 · 2018-03-03
**Contributions somewhat incremental**

**Rating:** 5
**Confidence:** 4

**Review:**

Thank you for an interesting read.

This paper proposed minimising the reverse KL for fitting implicit distributions. The reverse KL is estimated by GAN-based density-ratio estimation. Furthermore a generalisation of reverse KL that emphasises the entropy term is proposed. Experiments considered fitting the target distribution with two different set-ups: one with target distribution's samples only, and the other with tractable unnormalised density.

The paper is clearly written and easy to follow.

Although the experimental results look promising, I feel like the proposed approach is a straightforward combination of existing approaches:

1. GAN-based density ratio estimation is not new and the authors admit it. Probably the new part is the suggestion of extra entropy term (and its corresponding lower-bound), although I don't think it is a significant contribution.

2. Amortising dynamics is also not new and the authors also admit it. The new part is to replace the divergence that is used for matching to reverse KL (originally amortised SVGD uses L2, and amortised MCMC uses Jensen-shannon with GAN-based estimation), although again I wouldn't say it's highly novel.

However, the observation that amortised SVGD with L2 loss fails is interesting.

 In summary, I think the paper lacks significant novelty. Given that I don't understand the bar of acceptance for ICLR workshop, I will recommend for weak rejection, but I won't be offended if the abstract is accepted.

---

### Official Review · AnonReviewer4 · 2018-03-17
**Too squished**

**Rating:** 4
**Confidence:** 3

**Review:**

This paper has abused the format in an attempt to squash in too much material. It's really a long paper but without a proper introduction, or clarity of what the contribution is. There is previous work on mode collapse and amortized inference. Trying to cover both ideas in this amount of space doesn't work for me. I can't unpick what is actually new, or follow everything from the paper alone.

Minor: A large number of papers on variational methods use q for the approximating distribution and p for the target distribution. It seems gratuitously confusing to reverse that.

---

### Decision · Program_Chairs · 2018-03-20
**ICLR 2018 Workshop Acceptance Decision**

**Decision:**

Reject

**Comment:**

Based on the reviews, this paper has not been accepted for presentation at the ICLR workshop. However, the conversation and updates can continue to appear here on OpenReview.